# Visualizing the Functional Dynamics of P-Glycoprotein and Its Modulation by Elacridar via High-Speed Atomic Force Microscopy

**DOI:** 10.3390/ijms27010356

**Published:** 2025-12-29

**Authors:** Yui Kanaoka, Norie Hamaguchi-Suzuki, Yuto Nonaka, Soichi Yamashita, Osamu Miyashita, Atsuyuki Ito, Satoshi Ogasawara, Florence Tama, Takeshi Murata, Takayuki Uchihashi

**Affiliations:** 1Department of Physics, Graduate School of Science, Nagoya University, Furo-Cho, Chikusa-Ku, Nagoya 464-8602, Aichi, Japan; yui.kanaoka@d.phys.nagoya-u.ac.jp (Y.K.); florence.tama@nagoya-u.jp (F.T.); 2Department of Chemistry, Graduate School of Science, Chiba University, 1-33 Yayoi-Cho, Inage, Chiba 263-8522, Japanito-atsu38@chiba-u.jp (A.I.); satoshi-ogasawara@chiba-u.jp (S.O.); 3RIKEN Center for Computational Science, 6-7-1 Minatojima-Minamimachi, Chuo-Ku, Kobe 650-0047, Hyogo, Japan; osamu.miyashita@riken.jp; 4Center of Quantum Life Science for Structural Therapeutics (CQUEST), Chiba University, 1-33 Yayoi-Cho, Inage, Chiba 263-8522, Japan; 5Institute of Transformative Bio-Molecules, Nagoya University, Furo-Cho, Chikusa-Ku, Nagoya 464-8601, Aichi, Japan; 6Exploratory Research Center on Life and Living Systems (ExCELLS), National Institutes of Natural Sciences, 5-1 Higashiyama, Myodaiji, Okazaki 444-8787, Aichi, Japan; 7Institute for Glyco-Core Research (IGCORE), Nagoya University, Furo-Cho, Chikusa-Ku, Nagoya 464-8601, Aichi, Japan; 8Quantum-Based Frontier Research Hub for Industry Development (Q-BReD), Nagoya University, Nagoya 464-8601, Aichi, Japan

**Keywords:** P-glycoprotein, single-molecule dynamics, high-speed atomic force microscopy

## Abstract

P-glycoprotein (P-gp) is an ATP-driven transporter that effluxes a wide range of xenobiotics from cells, and its overexpression is a primary cause of multidrug resistance (MDR) in cancer. It is well-established that P-gp functions through conformational changes, yet its large-scale structural dynamics at work have been unexplored. Here, we directly visualized single P-gp molecules reconstituted in nanodiscs using high-speed atomic force microscopy (HS-AFM). The HS-AFM movies revealed that P-gp is intrinsically dynamic in its apo state, with its nucleotide-binding domains (NBDs) undergoing large, spontaneous opening and closing motions. However, addition of ATP stabilized a conformation characterized by NBD proximity with a strong tendency toward closure. We then leveraged this dynamic viewpoint to elucidate the relationship between Elacridar’s function and the resulting structural dynamics of P-gp. Elacridar is designed to overcome multidrug resistance (MDR) in cancer and acts as a potent dual inhibitor of both P-gp and the Breast Cancer Resistance Protein (BCRP), effectively blocking the drug efflux function of these transporters. This inhibitor has suggested concentration-dependent function: it is effluxed as a substrate at low concentrations and acts as an inhibitor at high concentrations. Our direct observations revealed that low concentrations induced active dynamics in P-gp, whereas high concentrations severely restricted its motion, leading to a rigid, non-productive state. Our study provides critical insights into how observing molecular motion itself can unravel complex biological mechanisms.

## 1. Introduction

P-glycoprotein (P-gp) is a well-characterized membrane transporter belonging to the ATP-binding cassette (ABC) family, which is predominantly expressed in the epithelial cells of the small intestine, kidney, liver, and brain [1,2,3,4,5]. Utilizing the energy derived from ATP hydrolysis, P-gp effluxes drugs and xenobiotics out of the cell, thereby playing a crucial role in biological defense mechanisms [2,6].

P-gp recognizes a broad range of substrates, primarily hydrophobic compounds, including various drugs, hormones, anticancer agents, and antibiotics [7,8,9,10,11,12]. However, this broad substrate specificity underlies the major clinical problem of drug resistance. In cancer cells, the overexpression of P-gp causes multidrug resistance (MDR) by effluxing anticancer agents before they reach effective intracellular concentrations, severely diminishing the efficacy of chemotherapy [13,14,15]. For instance, paclitaxel (Taxol), which inhibits microtubule depolymerization, is also a substrate of P-gp, and its therapeutic efficacy is consequently compromised by this efflux activity [16,17].

To overcome MDR, extensive research has aimed to both elucidate the transport mechanism of P-gp and develop effective inhibitors. These efforts, particularly through structural biology techniques like cryo-electron microscopy (cryo-EM), have provided invaluable insight into the static conformational information of the P-gp transport cycle [18] and how inhibitors bind [19,20,21]. However, P-gp is inherently a dynamic molecular machine [22], and these static snapshots fail to capture its functional dynamics—the real-time conformational changes that drive transport and inhibition. Consequently, this critical aspect of its mechanism has remained elusive.

This knowledge gap is particularly evident when considering complex modulators like Elacridar. As a prominent P-gp inhibitor, Elacridar is known to exhibit a concentration-dependent function, stimulating transport at low concentrations while inhibiting it at high concentrations [23,24]. A recent cryo-EM study revealed at least three distinct drug-binding pockets for Elacridar, suggesting that its concentration-dependent function arises from the specific number and combination of occupied sites [25]. However, because P-gp operates through a dynamic cycle of large conformational changes (e.g., inward- to outward-facing) [26], it remains unclear from these static models how the concentration-dependent occupancy of these multiple sites modulates the functional dynamics of P-gp. Thus, the precise mechanism of this complex modulator remains poorly understood, necessitating direct, real-time visualization of P-gp’s functional dynamics.

To address these challenges, we employed high-speed atomic force microscopy (HS-AFM) in this study, a technique unique in its ability to visualize the real-time, nanoscale structural changes in a single protein molecule under physiological conditions [27,28,29,30]. For instance, a previous study used this method to visualize the biogenesis of nascent high-density lipoprotein (HDL) mediated by another ABC transporter protein, ABCA1 [31]. Using P-gp reconstituted into a nanodisc, we directly observed its structural dynamics in both the presence and absence of ATP. Furthermore, by tracking the dynamic changes upon the addition of Elacridar, we sought to elucidate its complex inhibitory mechanism at the single-molecule level. Here, we demonstrate that P-gp undergoes spontaneous opening and closing motions, and that Elacridar switches its function by modulating these dynamics: stimulating the transport cycle at low concentrations while locking the transporter in a rigid, non-productive state at high concentrations.

## 2. Results

### 2.1. HS-AFM Observation of Nanodisc-Reconstituted P-gp in the Apo State

First, we performed HS-AFM observation of P-gp reconstituted into a nanodisc (P-gp-ND) in the nucleotide-free (apo) state. The HS-AFM images captured at 100–200 ms/frame showed individual P-gp-ND particles exhibiting two protrusions that repeatedly undergo dynamic opening and closing motions (Figure 1a, Appendix A). This bifurcated structure was confirmed to correspond to the pair of nucleotide-binding domains (NBDs) by comparison with cryo-EM images of the same sample (Appendix A).

To quantitatively evaluate the dynamics of this opening and closing motion, the ‘opening angle,’ defined by two points on the NBDs and one point on the nanodisc (as illustrated in Figure 1b), was measured in each frame of the AFM movies to generate a histogram (Figure 1b). Fitting this distribution with a Gaussian function yielded a median value of 47°, with the observed angles ranging from a minimum of 23° to a maximum of 111°, indicating a very broad distribution.

Next, to compare the conformational changes observed in the AFM images with known three-dimensional structures, we constructed models mimicking the nanodisc-reconstituted state by computationally adding an MSP ring and lipids to the existing closed (PDB: 6C0V) and open (PDB: 7OTI) structures of P-gp (Figure 1c,d). Simulated AFM images generated from these structural models [32] yielded opening angles of 20° for the closed structure and 55° for the open structure. To further validate our approach, we also analyzed the same sample by cryo-EM in the apo state (Appendix A). Simulated AFM images created from the resulting electron density maps showed two distinct conformations with angles of 24° and 42° (Appendix A). To quantify the effective spatial resolution, we compared the cross-sectional profiles of the NBD region in the experimental HS-AFM images with those of the simulated AFM images generated from the PDB models. The experimental profiles closely matched the simulated profiles when a low-pass filter corresponding to a 2 nm resolution was applied (Appendix A). Consequently, we estimate the actual spatial resolution achieved in our measurements to be approximately 2 nm. The opening angle histogram generated from the HS-AFM images reveals significantly larger opening angles compared to the results obtained from cryo-EM analysis, indicating a fundamental inconsistency between the static structures captured by cryo-EM and the dynamic conformations observed in solution by HS-AFM. This discrepancy is likely attributable to the methodological differences: while HS-AFM captures molecular movement in an aqueous, near-physiological environment, cryo-EM provides static snapshots under cryogenic conditions, which inherently restricts molecular movement. Furthermore, the large, highly open structure observed by HS-AFM likely represents a minor conformational subpopulation that would be excluded or averaged out during the cryo-EM reconstruction process. Additionally, cryo-EM studies often stabilize flexible domains using antibodies, nanobodies or removing native glycosylation to achieve high resolution, meaning the resulting structures are of a state where flexibility is artificially constrained. Conversely, the HS-AFM data, even in the absence of nucleotides, captures dynamic opening and closing motions involving large conformational changes, thereby demonstrating that P-glycoprotein exists in a highly flexible and fluctuating state in solution.

### 2.2. HS-AFM Observation of Nanodisc-Reconstituted P-gp in the Presence of ATP

First, to confirm that our P-gp-ND preparation was functionally active, we measured its ATPase activity at various ATP concentrations (Appendix A). The activity reached saturation at around 2 mM ATP, so this concentration was selected for subsequent HS-AFM experiments to investigate ATP-induced conformational changes. Contrary to the extensive dynamics observed in the apo state, the HS-AFM images resolved a conformation in which the bifurcated structure extending from the nanodisc adopted a seemingly closed state (Figure 2a, Appendix A). Due to the inherent spatial resolution limit of HS-AFM, it is not possible to definitively distinguish whether the state observed upon ATP addition represents a fully closed structure (outward-facing, closed structure with NBD binding by ATP) or merely a state where the NBDs are in close proximity. Additionally, we conducted observations in the presence of ATP and vanadate. However, even under these conditions, the inherent resolution limit of HS-AFM prevented us from distinguishing whether the NBDs were fully dimerized (ATP-bound) or merely in close proximity, as both states exhibit similar topological features. Nevertheless, it is unequivocally clear that the addition of ATP results in a significant shift away from the large-scale dynamics seen in the apo state, leading to the predominance of a conformation characterized by close NBD proximity or full NBD binding induced by ATP.

To further investigate the ATP concentration-dependent change, we measured the NBD opening angles at various ATP concentrations and evaluated the resulting distributions (representative data in Figure 2b; full data in Appendix A). We fitted each distribution with three Gaussian functions, fixing one peak at 47°, the median angle of the apo state. The molecular ensemble under ATP turnover conditions is considered to contain a fraction of molecules in the transient apo state. Therefore, fixing the peak position to that of the apo state determined in Figure 1b reduced the degrees of freedom, allowing us to more accurately isolate and characterize the emerging ATP-dependent states. This ATP-dependent change in the angle distribution revealed a progressive narrowing with increasing ATP concentration, indicating a clear trend toward the predominance of the seemingly closed state. A comparison of the fitted peak positions at each concentration showed that Peak 1 remained stable at approximately 20°, regardless of the ATP concentration. This Peak 1, which we define as the closed-biased state, is considered to correspond to a conformation characterized by close NBD proximity or full NBD binding induced by ATP, within the permissible spatial resolution of HS-AFM. The observation that the area under this distribution increases as the concentration rises strongly suggests that the closed-biased state becomes preferentially dominant in a concentration-dependent manner (Figure 2c upper and lower panel). In contrast, the center of Peak 2, which represents the intermediate state, exhibited a continuous shift to smaller angles with increasing ATP concentration, moving from 37° (at 0.1 mM) to 27° (at 2 mM). This indicates that this state is not a single fixed conformation but rather represents a dynamic ‘tightening’ process where the NBDs progressively approach each other before achieving the closed-biased state (Figure 2c upper panel). While the precise number of these intermediate states (corresponding to the number of fitting peaks) could not be definitively determined, the area under the distribution corresponding to this intermediate state increases in a concentration-dependent manner. Consequently, it is evident that the presence of ATP preferentially favors the closed-biased state, along with its associated intermediate states. Recent cryo-EM studies define the “occluded state” as a tightly dimerized conformation bound to ATP before transitioning to an outward-facing structure [33]. In our HS-AFM observations, this geometry corresponds to the closed-biased Peak 1. Consequently, the intermediate Peak 2 (37–27°) likely represents a transient state formed during the initial NBD docking process, capturing the dynamic transition path that precedes full dimerization. Furthermore, the fixed peak at 47° remained present across all tested ATP concentrations but showed a decrease in its distribution area corresponding to the increase in ATP concentration.

The conformational populations identifiable from the HS-AFM images were observed to shift quantitatively from the open state in apo state (47°) towards a closed-biased state (20°) and an intermediate state (transitioning from 37° to 27°). The fixed peak at 47° is assigned to the thermally fluctuating open state present in the apo condition. Crucially, the distribution areas corresponding to both the closed-biased state (20°) and the intermediate state were found to increase with rising ATP concentration. This observation suggests that increasing the ATP concentration suppresses the non-productive thermal fluctuation-driven motion (47°) and promotes the progression of the functional closed-biased and intermediate states that are believed to contribute to the ATP hydrolysis cycle.

### 2.3. HS-AFM Observation of Nanodisc-Reconstituted P-gp with the Inhibitor Elacridar

To investigate the effect of Elacridar, we performed HS-AFM observations of P-gp-ND in the presence of the inhibitor. In the absence of ATP, Elacridar alone induced a concentration-dependent stabilization of the closed-biased state. Increasing the Elacridar concentration caused a clear shift in the main peak of the NBD opening angle distribution toward smaller angles (representative data in Figure 3c; full data in Appendix A). For instance, at 5 μM Elacridar, the large opening and closing motions were suppressed, and the time spent in the closed-biased state increased significantly (Figure 3a, Appendix A). Remarkably, even at the very low concentration of 5 nM, Elacridar was observed to inhibit the dynamics of P-gp (representative data in Figure 3c; full data in Appendix A). This observation suggests that Elacridar possesses a high binding affinity for P-gp, allowing it to significantly alter the protein’s structural dynamics even without ATP.

In the presence of 2 mM ATP and 5 μM Elacridar, a similar tendency towards a stable, closed-biased state was observed, indicating that Elacridar promotes this conformation regardless of the presence of ATP (Figure 3b, Appendix A). Furthermore, varying the Elacridar concentration in the presence of ATP revealed its dual function at a dynamic level (representative data in Figure 3d; full data in Appendix A). At a low concentration of Elacridar (5 nM), the angle distribution remained broad, resembling the dynamics of the 2 mM ATP-only state. In contrast, high concentrations (500 nM and 5 μM) caused the distribution to collapse into a single, narrow peak at a small angle, indicating that the NBDs were locked in a tightly closed-biased state for inhibition. This concentration-dependent switch in dynamics provides a structural basis for the mechanism by which Elacridar transitions from a transported substrate to an inhibitor.

## 3. Discussion

### 3.1. Dynamics of P-gp-ND in the Nucleotide-Free Apo State

In this study, we provided the first direct, real-time visualization of the functional dynamics of a single P-gp molecule using HS-AFM. Our observation of P-gp-ND in the apo state revealed that the NBDs undergo repeated and pronounced opening and closing motions. The scale of this motion, which resulted in opening angles exceeding 80°, far surpassed what would be expected from previously reported static structures (inward facing, open structure, PDB: 7OTI). This discrepancy arises primarily because HS-AFM directly visualizes molecules in an aqueous solution, enabling measurement under conditions where molecular motion is unrestricted. Conversely, cryo-EM requires the sample to be flash-frozen, and often utilizes stabilizing agents like antibodies or nanobodies, or requires the removal native glycosylation to enhance resolution, thus resulting in observations of a molecular motion-constrained state. Furthermore, structures representing minor conformational populations are frequently excluded during the computational analysis inherent to cryo-EM. This strongly suggests that highly open conformations, such as the 80° opening angle observed by HS-AFM, are likely eliminated from the final reconstruction. Consequently, the HS-AFM data indicates that P-glycoprotein can exhibit far greater dynamics than has been previously recognized.

To verify whether such large conformational changes are structurally plausible without compromising the structural integrity of P-gp, we performed Normal Mode Analysis (NMA) [34] using the nucleotide-free structure (inward facing, open structure, PDB: 7OTI) as a model (Appendix A). This analysis allows for the identification of the most probable, large-scale conformational changes by calculating the low-frequency vibrational modes of the protein. The results showed that the NBD could open to approximately 80° in the simulation without compromising the overall protein structure (Appendix A). Moreover, the opening angle from the simulated AFM image generated from NMA structure (80°) falls within the distribution of large-scale dynamics observed by HS-AFM, although the experimental distribution extends further (up to 111°) likely due to stochastic thermal fluctuations exceeding the harmonic limits of NMA. To investigate the effect of the opening angle due to the molecular tilt relative to the substrate, we created a simulation image of the P-gp-ND structural model simulated by NMA while tilted relative to the substrate. The result showed an opening angle of 82°, indicating that the substrate tilt has little significant effect (Appendix A). These results indicate that the large opening and closing motion observed by HS-AFM is attributable to the intrinsic structural fluctuations of P-gp in the apo state.

In summary, our results revealed that the apo state of P-gp is not a single, static entity but a highly dynamic machine, constantly undergoing large structural fluctuations. The observed large amplitude opening in the apo state is likely driven by thermal fluctuations. This interpretation is supported by previous MD studies demonstrating that P-gp possesses intrinsic flexibility and samples a wide range of NBD distances [35]. This detailed view of the transporter’s ground-state dynamics, made possible by direct HS-AFM visualization in a solution environment, provides a crucial baseline for interpreting how its function is modulated by ligands such as ATP and drug-like inhibitors.

### 3.2. Further Performance for Dynamics of P-gp-ND in the Presence of ATP

In our analysis of the ATP-induced closed-biased state (Peak 1), we acknowledged that the distinction between simple NBD proximity and ATP-induced tight dimerization remains ambiguous due to the inherent spatial resolution of HS-AFM. Addressing the current spatial resolution limits represents a key challenge for future studies. Potential strategies to overcome this include the use of cantilevers with sharper tips, which would directly enhance physical resolution. Furthermore, computational post-processing techniques such as Localization AFM (LAFM) [36] offer a promising avenue. LAFM reconstructs high-resolution images by calculating the peak positions of fluctuating molecules over multiple frames. Although this approach requires stable, long-term HS-AFM imaging to accumulate a sufficient number of frames for high-resolution reconstruction, applying LAFM to the closed-biased state could potentially reveal the precise packing of the NBD dimer at a resolution surpassing the physical limitation of AFM.

### 3.3. Relationship Between P-gp Inhibition, Dynamics, and Activity

To elucidate Elacridar’s complex inhibitory mechanism, we next examined its direct impact on the structural dynamics of P-gp. In the absence of ATP, Elacridar alone induced a concentration-dependent shift in the NBD opening angle towards smaller values, indicating that increasing inhibitor concentration promotes a more closed-biased state (representative data in Figure 3c; full data in Appendix A). Furthermore, the shape of the distribution changed markedly between 50 nM and 500 nM, suggesting a potential change in the number of occupied binding sites within this concentration range. Notably, this effect was evident even at concentrations as low as 5 nM. The peak position in the distribution for 5 nM Elacridar alone is observed at 26° (Figure 3c). These findings demonstrate that Elacridar binds to P-gp with high affinity even at low concentration, stabilizing to closed-biased state and thereby directly impacting its structural dynamics even in the absence of ATP.

Under conditions containing 2 mM ATP and a low concentration of 5 nM Elacridar, the opening angle distribution was relatively broad, closely resembling that observed with 2 mM ATP alone (representative data in Figure 3d; full data in Appendix A). The similarity in the distribution shape to the condition solely performing ATP hydrolysis (2 mM ATP) is significant. The peak positions were also comparable: 20° and 28° for 2 mM ATP alone, and 21° and 31° for 2 mM ATP with 5 nM Elacridar. Crucially, the ATPase activity measurement under the 2 mM ATP and 5 nM Elacridar condition also revealed significantly high activity (Figure 4a). These findings collectively suggest that the P-gp complex adopts the necessary conformation for substrate extrusion, implying that Elacridar is being transported as a substrate. This result supports the prior hypothesis that at low concentrations, Elacridar initially binds to the P-gp pocket with the highest affinity and is subsequently actively extruded as a substrate [25].

In contrast, at high concentrations (≥500 nM), Elacridar functioned as an inhibitor, as evidenced by a marked reduction in ATPase activity (Figure 4a). Under these conditions, our HS-AFM data showed that the NBD opening angle distribution narrowed dramatically, with its peak shifting to a smaller angle (representative data in Figure 3d; full data in Appendix A). This closed-biased state likely occurs because Elacridar occupies multiple binding pockets, which suppresses the large-scale conformational changes required for the transport cycle. Previous reports have established that, for substrate efflux, the transmembrane (TM) helices must undergo conformational changes to enable the transition to an outward-facing conformation [20,37]. However, it has also been reported that when Elacridar binds to two or more pockets, it physically obstructs these TM helix movements through steric hindrance, thereby preventing the necessary conformational transition [20,25]. Integrating this knowledge with our observation that high Elacridar concentrations lock the NBDs in a closed-biased state, we conclude that P-gp is trapped in an inhibitory condition. In this case, the transporter is unable to transition to the outward-facing conformation required for efflux.

In summary, our results provide a single-molecule perspective on the dual function mechanism of Elacridar. We show that at low concentrations, it functions as a substrate that drives the transport cycle (Figure 4b, left). At high concentrations, however, it acts as an inhibitor that occupies multiple binding pockets, thereby arresting the structural dynamics and trapping the cycle (Figure 4b, right). Our work thus provides a clear structural and dynamic basis for Elacridar’s paradoxical concentration-dependent effects, bridging the gap between static structures and biochemical activity.

## 4. Materials and Methods

### 4.1. Purification of P-gp-ND

Human P-gp was purified and reconstituted into nanodiscs as previously described (19) with minor modifications. The P-gp coding sequence was cloned into a pEG vector bearing a C-terminal HRV3C protease site, mNeonGreen, and an 8 × His tag. Expi293F cells (obtained from Thermo Fisher Scientific, Waltham, MA, USA) were transfected with polyethyleneimine (PEI), cultured in HE200 CD medium, and harvested 72 h post-transfection; pellets were washed and stored at −80 °C. Cells were solubilized with n-dodecyl-β-D-maltoside (DDM)/cholesterol hydrogen succinate (CHS), and P-gp was captured on anti-His antibody resin. For on-resin reconstitution, bound P-gp was incubated with MSP1D1 and brain lipid/cholesterol (8:2, *w*/*w*) at a molar ratio of 1:10:350 (P-gp:MSP1D1:lipid) for 1 h at 4 °C, followed by detergent removal with Bio-Beads SM-2 (Bio-Rad, Hercules, CA, USA). The resin was washed with >10 column volumes of 25 mM HEPES (pH 7.5), 150 mM NaCl, and (reconstituted) P-gp-ND was released by HRV3C protease cleavage. The eluate was concentrated (100 kDa MWCO) and polished by Superdex 200 size-exclusion chromatography (Cytiva, Marlborough, MA, USA) in 25 mM HEPES (pH 7.5) and 150 mM NaCl. Peak fractions were pooled, concentrated, and analyzed by SDS-PAGE and size-exclusion chromatography (Appendix A) before being stored at −80 °C.

### 4.2. Measurement of ATPase Activity

ATPase activity of purified P-gp-ND was measured using an ATP-regenerating system [38]. ATP hydrolysis rates at 37 °C were determined based on NADH oxidation, monitored as a decrease in absorbance at 340 nm. The reaction mixture (200 µL) contained 2.5 mM phosphoenolpyruvate, 50 µg/mL pyruvate kinase, 50 µg/mL lactate dehydrogenase, 0.2 mM β-NADH (dipotassium salt), and 5 µg/mL P-gp in 25 mM HEPES (pH 7.5), 150 mM NaCl, 10 mM MgCl_2_. Various concentrations of Elacridar (1 nM–50 µM) were added, and the reaction was initiated by the addition of 2 mM ATP. After the reaction had stabilized, ATPase activity was measured at 30 s intervals for 30 min, and data from 5 to 30 min were used for analysis. Specific activities were calculated from three distinct 5 min intervals, and measurements were independently repeated three times. Mean values and standard deviations were calculated using GraphPad Prism 8 (GraphPad Software, San Diego, CA, USA).

### 4.3. Cryo-EM Sample Preparation and Data Collection

Grid preparation involved applying P-gp-ND (1.0 mg/mL) to glow-discharged Quantifoil R1.2/1.3 grids (Quantifoil Micro Tools GmbH, Großlöbichau, Germany) using a Vitrobot Mark IV (Thermo Fisher Scientific, Waltham, MA, USA). Vitrification was performed at 18 °C and 100% humidity, with a blotting time of 5 s and a blotting force of 10, followed by plunge-freezing in liquid ethane. Data acquisition was carried out on a 300 kV FEI Titan Krios microscope (Thermo Fisher Scientific, Waltham, MA, USA) equipped with a Gatan K3 Summit direct electron detector (Gatan, Inc., Pleasanton, CA, USA) operating in electron counting mode. A total of 8982 movies were collected at a physical pixel size of 0.75 Å (Appendix A). Each movie consisted of 49 frames recorded with a dose of 1.0 e^−^/Å^2^ per frame, over a defocus range of −0.8 to −2.0 μm (0.4 μm step).

### 4.4. EM Data Processing

The data processing workflow is summarized in (Appendix A). Image analysis was conducted using cryoSPARC version 4.6.2 (Structura Biotechnology Inc., Toronto, ON, Canada) [39,40]. Initial preprocessing steps included drift correction, beam-induced motion correction, and dose weighting, followed by patch CTF estimation. A total of 2,521,606 particles were picked using Blob Picker, extracted into 300 × 300 pixel boxes (binned 3 × 3), and subjected to 2D classification (K = 200). From this, 1,245,040 particles were selected for initial 3D model generation. The resulting ab initio maps served as references for heterogeneous refinement. The highest-resolution 3D class was selected for final 3D refinement. These particles were also subjected to 3D variability analysis [41] to visualize NBD motion (Appendix A). The final 3D refinements yielded maps with resolutions of 7.91 Å (open state) and 7.17 Å (closed state).

### 4.5. Sample Preparation for HS-AFM Observation

A mica substrate (1.5 mm in diameter, 0.1 mm thick; Furuuchi Chemical Co., Ltd., Tokyo, Japan) was mounted onto a cylindrical glass stage (2 mm in height) using adhesive glue. To adsorb the sample onto the mica substrate, a 2 µL aliquot of the P-gp-ND sample, diluted with observation buffer containing 25 mM HEPES (pH 7.0), 150 mM NaCl, was deposited onto the freshly cleaved mica and incubated for 5 min at room temperature. The substrate was then washed with 80 µL of observation buffer to remove any unadsorbed molecules. This procedure promotes the adsorption of the nanodiscs onto the mica surface in a side-on orientation [42], which allows the two NBDs of P-gp to be visualized from the side by HS-AFM. Additives such as ATP and Elacridar were pre-mixed with the sample to the desired final concentrations prior to deposition. For experiments involving ATP, the observation buffer also contained 2 mM MgCl_2_.

### 4.6. HS-AFM Observation and Instruments

All HS-AFM observations were performed at room temperature (25 °C) in tapping mode using a laboratory-built HS-AFM instrument [43]. The cantilever was oscillated at around its resonance frequency with a free amplitude of 1–2 nm. We used small cantilevers (BL-AC10DS-A2, Olympus Corporation, Tokyo, Japan) with the following specifications: 9 µm in length, 2 µm in width, 0.13 µm in thickness, a spring constant of ~0.1 N/m, a resonance frequency of 400–600 kHz in water, and a quality (Q) factor of ~2 in solution. The probe tip was prepared by growing an amorphous carbon tip on the original cantilever via electron-beam deposition (EBD) in a scanning electron microscope (SEM), followed by sharpening with argon gas plasma etching to a final radius of 2–5 nm.

### 4.7. Image Processing and Data Analysis of HS-AFM Data

HS-AFM images were processed using custom software based on Igor Pro 9 (WaveMetrics, Inc., Lake Oswego, OR, USA). To measure the opening angle of P-gp-ND, three points corresponding to the pixels with the highest intensity were selected in each frame: two points, one from each of the two NBD regions, and one from the nanodisc. The angle created by these three points was subsequently analyzed as the opening angle.

### 4.8. Generation of Simulated AFM Images

Simulated AFM images were produced by performing a collision simulation between a model AFM probe and the atomic coordinates of structures obtained from the PDB. To convert the electron density map (MRC file) obtained from cryo-EM into 3D coordinate data for the molecular structure, a surface extraction method using the Marching Cubes algorithm was employed. In this method, point cloud data representing the molecular surface was generated by extracting the isosurface at a defined threshold (isolevel = 0.3), which was then used to generate the simulated AFM image. Each structural model was oriented to match the scanning direction of the experimental setup. The AFM probe was modeled as a cone with a tip angle of 20° and a tip radius of 1.0 nm. To approximate the spatial resolution of the actual HS-AFM images, a low-pass filter with a cutoff spatial frequency of 0.5 nm^−1^ was applied to the simulated images. This cutoff frequency was determined by comparing the cross-sectional profiles of the experimental HS-AFM images with the simulated images, identifying the value that yielded the highest structural correlation (Appendix A). All procedures were performed using custom software developed in Igor Pro 9 (WaveMetrics, Inc., Lake Oswego, OR, USA).

### 4.9. Embedding and Modeling of Known Structures into Nanodiscs

We constructed atomic models of P-glycoprotein (P-gp) embedded in a nanodisc using the CHARMM-GUI version 3.8 Nanodisc Builder (https://charmm-gui.org). Two known P-gp structures were used as templates: the closed structure (PDB ID: 6C0V) and the open structure (PDB ID: 7OTI). In the builder, each structure was embedded into a model structure of a nanodisc composed of the MSP1D1 scaffold protein and POPC lipids.

### 4.10. Dynamics Simulation Using Normal Mode Analysis

Normal Mode Analysis (NMA) is useful for predicting the intrinsic, collective motions of biomolecules, such as domain movements or conformational changes, based on their structure alone [34,44]. Moreover, it was also recently introduced to interpret AFM data [45]. NMA was performed for P-gp in its nucleotide-free state (PDB ID: 7OTI). An Elastic Network Model (ENM) with a distance cutoff set to 8 Å and all-atom selection was used for the normal mode calculations for the rapid investigation of protein dynamics [46]. To simulate the characteristic opening and closing motion of the NBDs, the structure was displaced along the lowest-frequency normal mode derived from the analysis [47].

## 5. Conclusions

In this study, we employed HS-AFM to visualize the real-time structural dynamics of P-gp reconstituted into nanodiscs, providing a dynamic perspective that complements static structural biology. We demonstrated that P-gp is intrinsically flexible, exhibiting large-amplitude spontaneous opening and closing motions of the NBDs in the apo state. ATP binding shifts this equilibrium, suppressing non-productive thermal fluctuations and promoting a closed-biased state.

Crucially, our single-molecule analysis elucidated the structural dynamic basis for the paradoxical concentration-dependent effects of the inhibitor Elacridar. We revealed that while low concentrations of Elacridar induce active conformational cycling indicative of substrate transport, high concentrations lock the transporter into a closed-biased state, effectively inhibiting its function. This distinction clarifies how a single molecule can act as both a substrate and an inhibitor depending on the stoichiometry of binding.

The ability to directly observe and quantify modulation of transporter dynamics suggests that HS-AFM can offer a unique perspective for analyzing the dynamic effects of pharmacological compounds on transporter function.

## Figures and Tables

**Figure 1 ijms-27-00356-f001:**
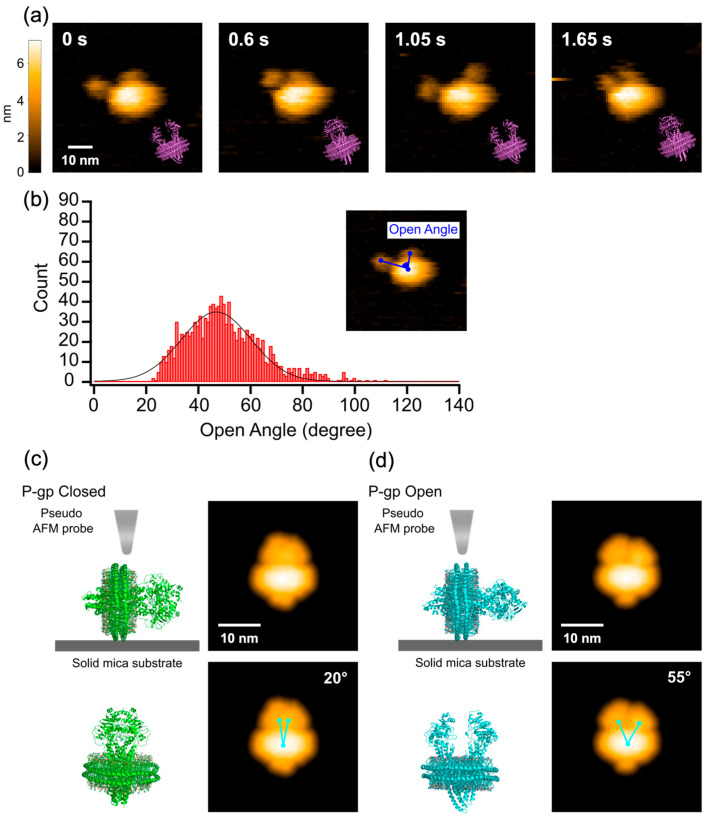
HS-AFM analysis of P-gp-ND in the apo state. (**a**) Representative HS-AFM snapshots of a single P-gp-ND in the apo state. The central bifurcated structure corresponds to the NBDs and exhibits an opening and closing motion. The purple insets show the corresponding structural models to indicate the molecular orientation. (**b**) Histogram of NBD opening angles measured from four independent molecules (*n* = 4, 1200 frames total). The black line represents a single Gaussian fit to the distribution, yielding a median angle of 47°. (**c**,**d**) Structural models of P-gp-ND and their corresponding simulated AFM images for the (**c**) closed (PDB: 6C0V) and (**d**) open (PDB: 7OTI) states by computationally embedding the structures in a nanodisc (MSP1D1 and POPC lipids). Simulated AFM images were generated with a probe tip radius of 1 nm and a cone angle of 20°. The opening angles were measured from these simulated images as indicated by the three light blue dots. See also Appendix A.

**Figure 2 ijms-27-00356-f002:**
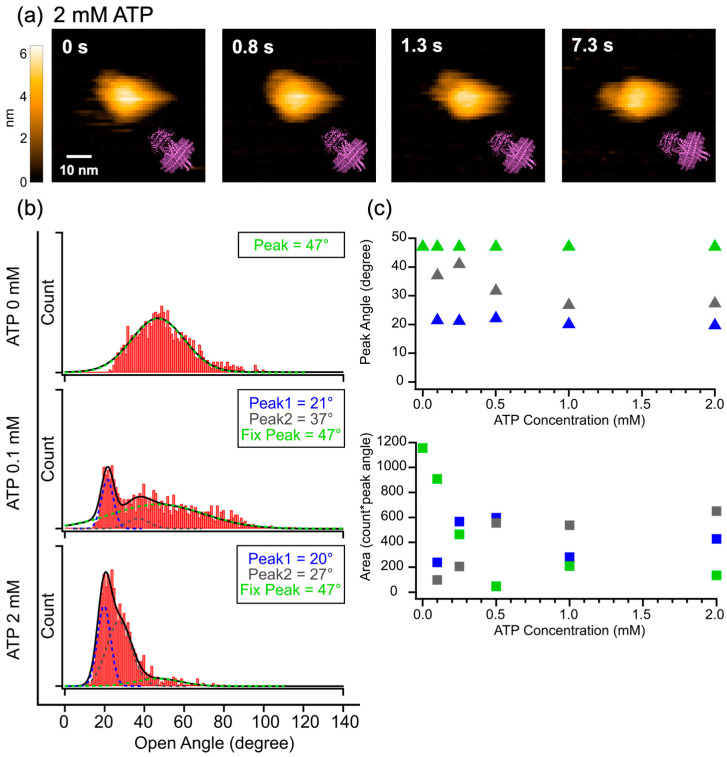
HS-AFM analysis of P-gp-ND in the presence of ATP. (**a**) Representative HS-AFM snapshots of a single P-gp-ND molecule in the presence of 2 mM ATP. The purple insets show the corresponding structural models to indicate the molecular orientation. (**b**) Representative histograms of NBD opening angles at the indicated ATP concentrations (ATP 0 mM, 0.1 mM, 2 mM). Each histogram was constructed from 1200 frames collected from multiple P-gp-ND molecules. The solid black lines represent the multi-peak Gaussian fit, and the dashed lines represent the individual Gaussian components. (**c**) (**upper panel**) NBD opening angles of the fitted Gaussian peaks plotted as a function of ATP concentration. (**lower panel**) Relative area of each peak plotted as a function of ATP concentration. In the upper panel, data points are shown as triangles, while in the lower panel, they are shown as squares. In both panels, colors correspond to the fixed peak (green), Peak 2 (gray), and Peak 1 (blue) from the distributions in (**b**) and Appendix A. See also Appendix A.

**Figure 3 ijms-27-00356-f003:**
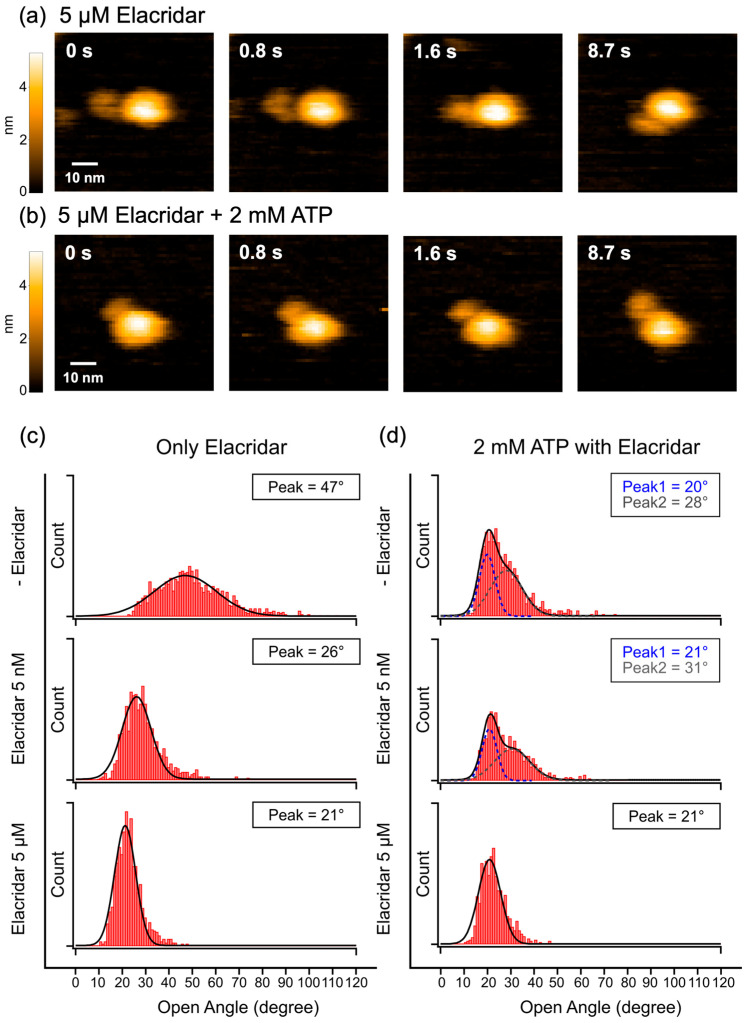
HS-AFM analysis of P-gp-ND with varying Elacridar concentrations. (**a**,**b**) Representative HS-AFM snapshots of a single P-gp-ND molecule in the presence of 5 μM Elacridar (**a**) without ATP and (**b**) with 2 mM ATP. (**c**) Representative histograms of NBD opening angles in the absence of ATP at the indicated Elacridar concentrations (Elacridar 0 nM, 5 nM, 5 μM). Each histogram was constructed from the analysis of 1200 frames. The solid black lines represent single Gaussian fits. (**d**) Representative histograms of NBD opening angles in the presence of 2 mM ATP at the indicated Elacridar concentrations (Elacridar 0 nM, 5 nM, 5 μM). The histogram for the control (0 nM Elacridar) was constructed from 1200 frames, while those in the presence of Elacridar were constructed from 900 frames each. The solid black lines represent the multi-peak Gaussian fits, and the dashed lines represent the individual Gaussian components. See also Appendix A.

**Figure 4 ijms-27-00356-f004:**
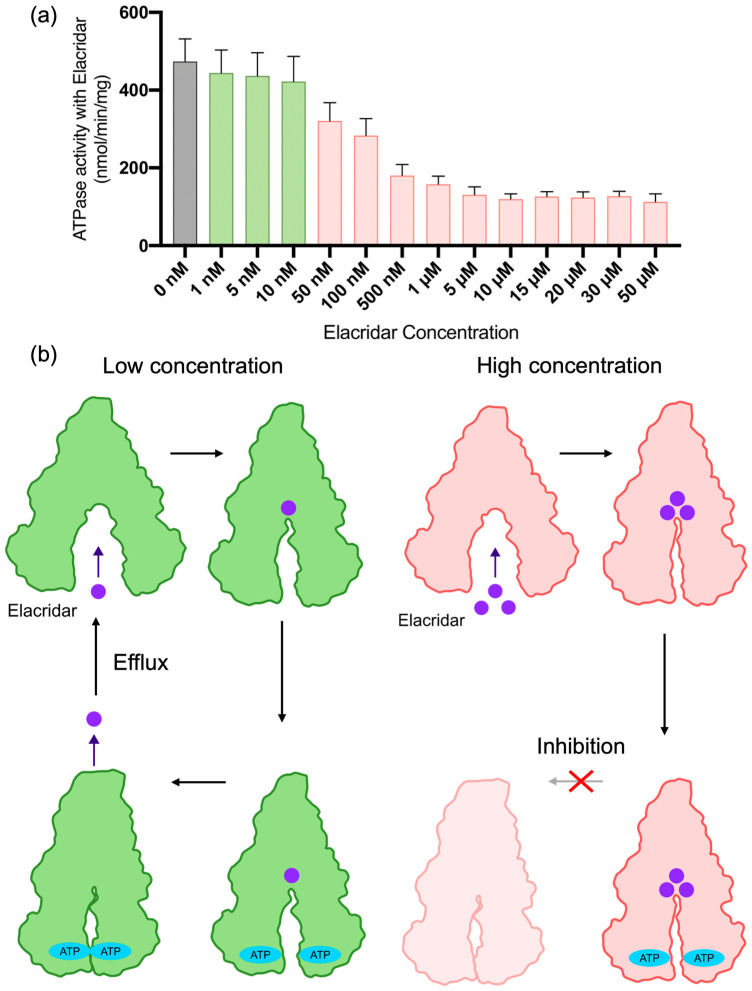
ATPase activity results at varying Elacridar concentrations and the proposed efflux-inhibition switching mechanism. (**a**) ATPase activity of P-gp-ND was measured in the presence of various concentrations of Elacridar. Data points represent the mean ± standard deviation (SD) of three experiments (*n* = 3). The bar colors distinguish between the stimulation phase (low concentrations, [green]) and the inhibition phase (high concentrations, [pink]). (**b**) A schematic model illustrating the proposed mechanism for how Elacridar switches from a substrate to an inhibitor. The purple circles represent Elacridar molecules. At low concentrations (**left**), Elacridar functions as a substrate, driving the transport cycle. At high concentrations (**right**), it acts as an inhibitor by occupying multiple binding pockets, which arrests structural dynamics and traps the transporter in an occluded state.

## Data Availability

The original data presented in the study are openly available at https://doi.org/10.5281/zenodo.17773030.

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
