# Peer review of "Visualizing the Functional Dynamics of P-Glycoprotein and Its Modulation by Elacridar via High-Speed Atomic Force Microscopy"

_ijms, 2025, doi:10.3390/ijms27010356_

Round 1
Reviewer 1 Report
Comments and Suggestions for Authors
This manuscript presents a compelling study utilizing high-speed atomic force microscopy (HS-AFM) to investigate the real-time structural dynamics of P-glycoprotein (P-gp) in nanodiscs. The authors provide direct visual evidence of nucleotide-binding domain (NBD) motions in the apo state, ATP-induced stabilization of closed conformations, and concentration-dependent modulation by Elacridar. The work is innovative, technically sound, and offers significant insights into the dynamic behavior of P-gp, particularly in elucidating the mechanism of Elacridar's dual functionality. The study is merits publication after minor revisions.
Concerns and Suggestions for Improvement
- Spatial Resolution Limitations: While HS-AFM captures dynamics effectively, the resolution limits to distinguish between fully closed NBD dimers and proximal states are acknowledged. The authors should briefly discuss how future technical improvements (e.g., sharper probes) could address this.
- Sample Size Justification: The analysis of "n=4 molecules" for apo-state dynamics (Figure 1b) seems limited. Clarify whether this refers to independent molecules or repeated measurements, and justify the statistical robustness (e.g., via reproducibility across samples).
- Intermediate States in ATP Titration: The assignment of intermediate states (Peak 2 in Figure 2c) could be further supported by referencing known structural intermediates (e.g., occluded states) from literature. A brief discussion would enhance interpretability.
Author Response
Comment1
Spatial Resolution Limitations: While HS-AFM captures dynamics effectively, the resolution limits to distinguish between fully closed NBD dimers and proximal states are acknowledged. The authors should briefly discuss how future technical improvements (e.g., sharper probes) could address this.
Reply1
We appreciate the reviewer’s constructive suggestion regarding future technical improvements. In the revised Discussion section, we have added a paragraph discussing potential strategies to overcome current resolution limits. Specifically, we mention two key approaches:
- Hardware improvements: As suggested, the use of cantilevers with sharper tips (radius ≦ 1 nm) would directly enhance spatial resolution.
- Post-processing techniques: We highlighted the potential of Localization AFM (LAFM) (Heath et al., Nature 594, 385–390 (2021)). Since LAFM reconstructs high-resolution images by calculating the peak positions of fluctuating molecules over multiple stable frames, it could potentially distinguish "stable proximity" from "tight dimerization" by analyzing the positional distribution of the NBDs from stable, long-term HS-AFM observations.
Comment2
Sample Size Justification: The analysis of "n=4 molecules" for apo-state dynamics (Figure 1b) seems limited. Clarify whether this refers to independent molecules or repeated measurements, and justify the statistical robustness (e.g., via reproducibility across samples).
Reply2
Regarding the reviewer’s point, we clarify that n = 4 refers to four independent molecules, not repeated measurements. In single-molecule analysis, the statistical population is determined by the number of conformational snapshots. We analyzed a total of 1,200 frames from these molecules. This dataset provides a sufficient population to construct a robust histogram. Crucially, we confirmed that the broad angle distribution was highly reproducible across all four independent molecules, ensuring that the observed dynamics are an intrinsic property of P-gp rather than an artifact of a specific molecule. We have revised the legend of Figure 1 to explicitly state " Histogram of NBD opening angles measured from four independent molecules (n = 4, 1200 frames total) ".)
Comment3
Intermediate States in ATP Titration: The assignment of intermediate states (Peak 2 in Figure 2c) could be further supported by referencing known structural intermediates (e.g., occluded states) from literature. A brief discussion would enhance interpretability.
Reply3
We have carefully examined recent literature, including the cryo-EM study describing the occluded conformation (Culbertson et al., Nat. Commun. 16, 3619 (2025)). According to the literature, the "occluded state" is characterized by tight NBD dimerization with bound ATP, structurally resembling the fully closed outward-facing conformation. In our HS-AFM analysis, such a tightly closed geometry (~ 20°) corresponds to Peak 1 (the closed-biased state) rather than Peak 2. Therefore, we interpret our states as follows:
- Peak 1 (~ 20°): We define this peak in the main text as the “closed-biased state”. This state likely encompasses both the functional “occluded state “and the “outward-facing closed state” described in cryo-EM studies, where the NBDs are fully dimerized.
- Peak 2 (37° ~ 27°): We propose that Peak 2 represents a transient intermediate preceding tight NBD dimerization. The shift from 37° to 27° with increasing ATP concentration likely reflects the dynamic process of ATP binding and the initial docking of NBDs before they lock into the rigid occluded state or full dimerized state (Peak 1).
We have added a brief discussion to clarify this interpretation in the revised manuscript citing these structural studies to clarify that our dynamic observations.
Reviewer 2 Report
Comments and Suggestions for Authors
This manuscript aims to establish the large-scale structural dynamics of P-Glycoprotein in its functional state. It provides a technical comparison and a comprehensive examination of the functional dynamics of P-Glycoprotein, particularly focusing on conformational changes during inhibition. The topic is interesting, and the technical comparison is effective. I recommend this manuscript for publication, with a few suggestions for the authors:
- Although the current purification and reconstitution of human P-Glycoprotein into nanodiscs is clear, I recommend performing quantitative analysis and presenting the images.
- The authors demonstrate a fundamental inconsistency in P-Glycoprotein conformations between the static structures captured by cryo-EM and the dynamic conformations observed in solution by HS-AFM. From this perspective, consider that posttranslational modifications, such as glycosylation, may also affect protein conformations, especially since P-Glycoprotein expressed in Expi293F cells undergoes glycosylation.
Author Response
Comment1
Although the current purification and reconstitution of human P-Glycoprotein into nanodiscs is clear, I recommend performing quantitative analysis and presenting the images.
Reply1
We appreciate the reviewer’s suggestion to include quantitative data on sample quality. In response, we have added the size exclusion chromatography (SEC) elution profile and SDS-PAGE gel images to the revised Supplementary material (Figure S9). These results quantitatively demonstrate the high purity of the purified P-gp protein and the monodispersity of the reconstituted P-gp nanodiscs. We have updated the Materials and Methods section to describe these characterization steps.
Comment2
The authors demonstrate a fundamental inconsistency in P-Glycoprotein conformations between the static structures captured by cryo-EM and the dynamic conformations observed in solution by HS-AFM. From this perspective, consider that posttranslational modifications, such as glycosylation, may also affect protein conformations, especially since P-Glycoprotein expressed in Expi293F cells undergoes glycosylation.
Reply2
We thank the reviewer for this insightful comment regarding post-translational modifications. As pointed out, the P-gp used in our HS-AFM study was expressed in mammalian Expi293F cells and retain its native glycosylation. We agree that these native modifications, which are fully preserved in our experiments, may significantly influence the protein’s conformational landscape. This stands in contrast to many structural studies (such as crystallography or cryo-EM), where glycosylation chains are often removed or mutated to reduce heterogeneity and improve resolution. Therefore, the presence of native glycosylation is likely a key factor contributing to the "flexible and dynamic" nature observed in our HS-AFM movies, distinguishing them from static structures. We have added a discussion in the revised manuscript, to explicitly state that the preservation of glycosylation may account for the conformational differences between our dynamic solution data and previously reported static structures.
Reviewer 3 Report
Comments and Suggestions for Authors
I have read the article Visualizing the Functional Dynamics of P-glycoprotein and Its Modulation by Elacridar via High-Speed Atomic Force Microscopy by Yui Kanaoka et al.
All in all this manuscript is written in a good manner, consists of solid valuable data and studies an important gap in structural biology linked with glycoprotein.
Authors could enhance their work, in my opinion, by:
-
Quantify the actual resolution achieved in nanometres for the flexible NBD regions.
-
Discuss In more detail, the mismatch between the maximum angle predicted by NMA and experimentally observed shows the model’s accuracy.
-
More accurate analytical justification about the ATP-bound state would be beneficial. Discuss why you can fix a Gaussian peak (the apo peak) during curve fitting.
-
Line 278: Change “In summarize…” to “In summary...".
-
It would be easier for the reader to see statistical methods in figures' notes.
-
Even a rough approximation of forces involved in conformational changes of P-gp would be beneficial. Authors can employ various approaches to assess these factors, such as molecular dynamics simulations, optical techniques using labelled Elacridar, or genetic engineering methods.
However, all of the above are only recommendations.
Author Response
Comment1
Quantify the actual resolution achieved in nanometres for the flexible NBD regions.
Reply1
Strictly speaking, quantifying the exact spatial resolution of AFM images is inherently challenging. This is because the achieved resolution is determined by a complex combination of factors, including the tip radius, the macroscopic geometry of the probe (tip aspect ratio), noise from the optical lever detection system, and the topography of the sample itself (e.g., flat surfaces vs. large vertical corrugations).
To address this, we attempted to quantify the resolution by evaluating how well the fine structural details of the nanodisc's lateral side were resolved. Specifically, we generated an "ideal" high-resolution image via collision simulation using a sharp conical probe with a tip radius of 0.5 nm. We then applied low-pass filters of varying spatial frequencies to this ideal image to simulate the resolution-limiting factors.
We found that the simulated image processed with a 2 nm cutoff low-pass filter exhibited a profile shape that most closely resembled the experimental AFM image (as shown in the profile comparison figure). Consequently, we estimate the effective spatial resolution of our measurements to be approximately 2 nm. This value is consistent with the spatial resolution typically achieved in previous HS-AFM studies (e.g., ~2 nm as reported in Uchihashi et al., Science 2011). We have clarified this explanation in the revised Methods and Results sections and showing this figure as Figure S3.

Comment2
Discuss In more detail, the mismatch between the maximum angle predicted by NMA and experimentally observed shows the model’s accuracy.
Reply2
We appreciate the reviewer’s comment regarding the quantitative comparison between simulation and experiment. We acknowledge the difference in absolute values; however, it is important to note that the experimental maximum of 111° represents an extreme tail of the distribution resulting from rare, large thermal fluctuations. In contrast, NMA identifies the most energetically accessible collective motions. The NMA prediction of ~80° opening is structurally significant and falls well within the upper range of our experimental distribution. Thus, we conclude that the NMA model successfully captures the directionality and intrinsic flexibility of the NBD opening motion, even if it does not reach the rare extremes of thermal fluctuation. We have clarified this interpretation in the revised Discussion.
Comment3
More accurate analytical justification about the ATP-bound state would be beneficial. Discuss why you can fix a Gaussian peak (the apo peak) during curve fitting.
Reply3
The rationale for fixing the Apo peak is based on the catalytic cycle. Under ATP turnover conditions, the molecular ensemble is a dynamic mixture that naturally includes a fraction of molecules in the transient nucleotide-free (Apo) state. By fixing the peak position to the value independently determined from the Apo-only condition (Fig. 1b), we reduced the degrees of freedom in the fitting process. This constraint prevents overfitting and allows for a more accurate and physically meaningful separation of the newly emerging ATP-bound/intermediate states from the background Apo population. We have added this analytical justification to the Methods section.
Comment4
Line 278: Change “In summarize…” to “In summary...".
Reply4
We have corrected the phrase to "In summary" as suggested.
Comment5
It would be easier for the reader to see statistical methods in figures' notes.
Reply5
We have updated the figure legends to include details of the statistical methods and sample sizes for better readability.
Comment6
Even a rough approximation of forces involved in conformational changes of P-gp would be beneficial. Authors can employ various approaches to assess these factors, such as molecular dynamics simulations, optical techniques using labelled Elacridar, or genetic engineering methods.
Reply6
We thank the reviewer for the suggestion to assess the forces involved. While direct force measurement is beyond the scope of this study, In the manuscript, we attribute the large-amplitude opening observed in the apo state to intrinsic thermal fluctuations. This interpretation is supported by previous MD simulations (Wen et al., J. Biol. Chem. 288, 19211–19220 (2013)), which demonstrated that P-gp possesses high intrinsic flexibility and samples a wide range of NBD distances in the inward-facing state without external force. We have expanded the Discussion section to cite this literature and explain that the observed dynamics are driven by thermal energy.
Reviewer 4 Report
Comments and Suggestions for Authors
The author delivered a very interesting method in studying dynamic enzymology process by using HS-AFM. I have the following comments:
- Other than the closed and open state of the transporter, the HS-AFM does not seem to provide other information. In other words, the single molecule technology should provide more detail information other than assemble behavior. This information seems not emphasized.
- There seems to be no clear connection between the open and closed state observed by addition of Elacridar with proposed model. If there is clear connection, please elaborate more in the paper. Some key experiments seem to be omitted. For example, what will happen if non-hydrolyzable ATP is used in the study? What will happen if Elacridar binding sites are mutated? If Elacridar is serving as a substrate, why there is no increase in terms of ATP hydrolysis?
Author Response
Comment1
Other than the closed and open state of the transporter, the HS-AFM does not seem to provide other information. In other words, the single molecule technology should provide more detail information other than assemble behavior. This information seems not emphasized.
Reply1
We appreciate the reviewer’s comment. We respectfully disagree that HS-AFM reveals only binary open/closed states. A unique advantage of HS-AFM is its ability to visualize the continuous conformational distribution and transition pathways that are inaccessible to static structural methods. Specifically, our histogram analysis revealed that the intermediate state (Peak 2) is not static a static conformation but exhibits a continuous shift from 37° to 27° as ATP concentration increases. We argue that this continuous shift provides critical dynamic information regarding the progressive tightening process of the NBDs prior to full ATP hydrolysis. This captured "motion" confirms that the transport cycle is driven by a flexible landscape rather than simple switching between rigid binary structures.
Comment2
There seems to be no clear connection between the open and closed state observed by addition of Elacridar with proposed model. If there is clear connection, please elaborate more in the paper. Some key experiments seem to be omitted. For example, what will happen if non-hydrolyzable ATP is used in the study? What will happen if Elacridar binding sites are mutated? If Elacridar is serving as a substrate, why there is no increase in terms of ATP hydrolysis?
Reply2
Regarding the additional experiments (non-hydrolyzable ATP or mutants), while they are interesting, the primary focus of this study is to elucidate the physical dynamics of the wild-type protein under physiological hydrolysis conditions.
Regarding the connection between the model and ATPase activity, we clarify that P-gp exhibits high intrinsic basal activity even in the absence of Elacridar. As shown in Figure 4a, P-gp maintains this high activity at low Elacridar concentrations (up to 10 nM), which supports the interpretation that Elacridar acts as a transported substrate in this range. Therefore, the hallmark of substrate recognition in this context is not the further stimulation of activity (which is already near maximal), but the maintenance of high turnover. In contrast, the dramatic inhibition of ATPase activity at high concentrations correlates directly with structural trapping into the closed-biased state observed in HS-AFM. This distinct switch from "high-turnover transport" to "structural arrest" strongly supports our proposed dual-mode mechanism.
Reviewer 5 Report
Comments and Suggestions for Authors
Minor Revision
This is an exciting, high-impact study (recommended for publication after minor revisions) that elevates HS-AFM from a visualization tool to a functional probe. It substantiates claims with multimodal data, challenging static views of P-gp and offering a dynamic framework for modulator design in MDR therapy.
Please address the following concerns:
1] Spatial Resolution and Conformational States: The paper acknowledges the spatial resolution limit (~1–2 nm) and how it affects the distinction between "closed-biased" (NBD proximity) and truly "closed" (dimerized) states (lines 150–157). To strengthen the claims regarding the "strong tendency toward closure," it would be highly beneficial to include detailed data from the ATP-vanadate experiments designed to trap the pre-hydrolysis state.
2] Apo State Flexibility and Functionality: While the description of the apo state’s "highly flexible" nature is compelling, the link to "non-productive thermal fluctuation-driven motion" suppressed by ATP (lines 191–194) implies functionality without direct supporting evidence. To better correlate dynamics with activity, integrating transport assays (e.g., verapamil efflux) under the High-Speed Atomic Force Microscopy (HS-AFM) conditions would be necessary.
3] Integrating ATPase Activity and Transport: Figure S5 confirms the ATPase activity of the complex, but it does not provide evidence of transport function. I recommend integrating a coupled transport assay (e.g., one utilizing relevant substrates) to directly link the observed structure and dynamics to functional transport activity.

Author Response
Comment1
Spatial Resolution and Conformational States: The paper acknowledges the spatial resolution limit (~1–2 nm) and how it affects the distinction between "closed-biased" (NBD proximity) and truly "closed" (dimerized) states (lines 150–157). To strengthen the claims regarding the "strong tendency toward closure," it would be highly beneficial to include detailed data from the ATP-vanadate experiments designed to trap the pre-hydrolysis state.
Reply1
We appreciate the reviewer’s suggestion. We actually performed preliminary HS-AFM observations using ATP-vanadate to trap the pre-hydrolysis state. However, we found that the conformation of the vanadate-trapped state was topologically indistinguishable from the "closed-biased" state (Peak 1) observed under ATP turnover conditions. Both exhibited the same NBD proximity angle of approximately 20°. Since the distinction between "tight proximity" and "full dimerization" is beyond the spatial resolution of HS-AFM (~2 nm), adding the vanadate data does not resolve this specific ambiguity. However, we emphasize that our claim of a "strong tendency toward closure does not rely on a static comparison with a trapped state, but is statistically substantiated by the dynamic population shift. As shown in Figure 2c, the significant, concentration-dependent increase in the area of Peak 1 provides robust quantitative evidence that the conformational equilibrium strongly shifts toward the closed-biased state in the presence of ATP.
Comment2
Apo State Flexibility and Functionality: While the description of the apo state’s "highly flexible" nature is compelling, the link to "non-productive thermal fluctuation-driven motion" suppressed by ATP (lines 191–194) implies functionality without direct supporting evidence. To better correlate dynamics with activity, integrating transport assays (e.g., verapamil efflux) under the High-Speed Atomic Force Microscopy (HS-AFM) conditions would be necessary.
Reply2
We thank the reviewer for the suggestion to correlate dynamics with transport activity. However, we must clarify a fundamental structural constraint: performing transport (efflux) assays is physically impossible in the nanodisc system. Unlike liposomes or whole cells, a nanodisc is a discoidal lipid bilayer wrapped by membrane scaffold proteins and lacks an internal compartment (a "closed" lumen). Consequently, there is no "inside" space to accumulate substrates, making it impossible to measure transport or efflux. Therefore, we assessed functionality using ATPase activity, which is the standard and widely accepted method for verifying the function of transporters reconstituted in non-compartmentalized systems like nanodiscs. As shown in Figure S6 (and Fig. 4a), our samples exhibited robust, ligand-responsive ATPase activity, confirming the protein is functional. Regarding the term "non-productive," we use it to describe the thermodynamic reality that the large fluctuations observed in the apo state are driven by thermal Brownian motion without energy input (ATP hydrolysis). Thus, this motion is energetically distinct from the directional, productive conformational changes that drive the transport cycle.
Comment3
Integrating ATPase Activity and Transport: Figure S5 confirms the ATPase activity of the complex, but it does not provide evidence of transport function. I recommend integrating a coupled transport assay (e.g., one utilizing relevant substrates) to directly link the observed structure and dynamics to functional transport activity.
Reply3
As detailed in our response to Comment 2, integrating a coupled transport assay is structurally unfeasible because nanodiscs do not possess an internal compartment to retain transported substrates. Instead, we rely on the ligand-stimulated ATPase activity (Figure 4a and Figure S6). The data clearly demonstrate that the P-gp-nanodisc complex retains high basal activity and responds to Elacridar in a concentration-dependent manner (substrate-like behavior at low concentrations and inhibition at high concentrations). This ligand-responsive ATPase profile serves as the definitive biochemical evidence linking the observed structures to functional activity in our study.
Round 2
Reviewer 4 Report
Comments and Suggestions for Authors
All my suggestions are address.